# Perceived Stress and Increased Food Consumption during the ‘Third Wave’ of the COVID-19 Pandemic in Spain

**DOI:** 10.3390/nu13072380

**Published:** 2021-07-12

**Authors:** Eduardo Sánchez-Sánchez, Jara Díaz-Jimenez, Ignacio Rosety, Maria José M. Alférez, Antonio Jesús Díaz, Miguel Angel Rosety, Francisco Javier Ordonez, Manuel Rosety-Rodriguez

**Affiliations:** 1Internal Medicine Department, Punta de Europa Hospital, Algeciras, 11207 Cádiz, Spain; 2Instituto de Investigación e Innovación Biomédica de Cádiz (INiBICA), Hospital Universitario Puerta del Mar, Universidad de Cádiz, 11009 Cádiz, Spain; 3Campus Cádiz, Doctoral School of the University of Cádiz (EDUCA), Edificio Hospital Real (PrimeraPlanta), Plaza Falla 8, 11003 Cádiz, Spain; luna_nueva17@hotmail.com; 4Human Anatomy, School of Medicine, University of Cádiz, Plaza Fragela, s/n, 11003 Cadiz, Spain; Ignacio.rosety@uca.es (I.R.); franciscojavier.ordonez@uca.es (F.J.O.); 5Department of Physiology, Faculty of Pharmacy, Campus Universitario de Cartuja, University of Granada, 18011 Granada, Spain; malferez@ugr.es; 6Medicine Department, School of Nursing, University of Cadiz, Plaza Fragela, s/n, 11003 Cadiz, Spain; antoniojesus.diaz@uca.es; 7Move-It Research Group, Biomedical Research and Innovation Institute of Cadiz, Puerta del Mar University Hospital, University of Cádiz, Plaza Fragela, s/n, 11003 Cadiz, Spain; Miguelangel.rosety@uca.es; 8Medicine Department, School of Medicine, University of Cádiz, Plaza Fragela, s/n, 11003 Cadiz, Spain; manuel.rosetyrodriguez@uca.es

**Keywords:** COVID-19 pandemic, food consumption, perceived stress, Perceived Stress Scale

## Abstract

The COVID-19 pandemic has impacted the mental health of people worldwide. An increase in perceived stress can lead to unhealthy behaviors such as increased food consumption. The aim of this study was to find the level of perceived stress and its relationship with increased food consumption during the “third wave” of the COVID-19 pandemic in Spain. This was a cross-sectional study that employed anonline self-reported frequency of consumption questionnaire and the Perceived Stress Scale-10. A total of 637 subjects participated and 83.6% of respondents had moderate or high stress—more prevalent in the female and young respondents. Moreover, 36.1% of respondents reported that they had increased the frequency of consumption of some foods, mainly nuts, snacks, and jellybeans, along with coffee, tea, cocoa, and soft drinks. Eating between meals was more pronounced in those with high stress (65.1%) than in those with moderate stress (40.4%) and low stress (20.2%). Furthermore, the respondents with high stress reported greater weight gain. Thus, the results show that the level of perceived stress during the ‘third wave’ of this pandemic increased food consumption.

## 1. Introduction

In December 2019, a series of atypical pneumonia cases were reported in Wuhan, China, caused by a new virus, which spread rapidly and had high infectivity and morbidity among humans. Subsequently, the World Health Organization had temporarily named this virus as a new coronavirus 2019 (2019-nCoV), and the disease it caused as COVID-19. Shortly afterward, it declared the situation a global public health emergency [1,2,3].

In Spain, the situation has been volatile: as soon as things start to stabilize, a new round of infection is reported, sometimes worse than the previous one. This European country is currently battling a “third wave” that has increased the number of deaths as well as the pressure on healthcare. On 19 February 2021, official data showed that the number of cases worldwide stood at 109,594,835, of which 3,133,122 were from Spain [4].

This has led many countries to implement different containment measures to reduce the spread of the virus [5]—social distancing, restriction of movement and the use of masks—which, together with the fear of contagion, concern for the safety of oneself or one’s loved ones, and the economic problems derived from these measures, has caused people to face previously unknown psychosocial situations. This has increased the risk of mental health problems such as anxiety, depression, and stress [6,7,8,9]. Several longitudinal studies have already shown that national “lockdowns” during the early phases of the pandemic had a negative impact on people’s mental health and subjective well-being [10,11,12,13].

Perceived stress can affect people’s mental health in different ways [14]: it is a person’s ability to interpret life events as overwhelming, erratic, or uncontrollable. Perceived stress during the COVID-19 pandemic emphasizes the extent to which an individual believes he or she has control over unexpected or difficult events or emotions arising from the situation (illness, isolation, and/or financial problems) [15]. Duration of measures, frustration, fatigue, inadequate information, economic problems, fear of contagion, media reports, etc., are considered as stressors that increase the stress perceived by individuals [16]. Stress experienced as a consequence of the pandemic is related to health and quality of life problems as well as to the emergence of less healthy habits such as sedentary lifestyles, increased exposure to digital screens, substance abuse such as alcohol or food addiction, etc. [17,18].

Addictive behaviors include binge eating without subsequent compensation, continuous eating, even in small amounts (snacking), and the consumption of foods high in calories, sugars and fats, which can interfere with the limbic system of the brain. These behaviors, which can lead to an increase in body weight and thus body mass index (BMI), fall under the umbrella of “emotional eating” (i.e., the consumption of food in response to negative emotions without following hunger cues) [19,20,21,22].

Perceived stress may play an important role in the increased food consumption and the consumption of foods considered less healthy. Therefore, the aim of this study was to find the level of perceived stress and its relationship with increased food consumption during the “third wave” of the COVID-19 pandemic in Spain.

## 2. Materials and Methods 

### 2.1. Study Design and Participants

A descriptive cross-sectional study was carried out based on an online questionnaire using a web platform, where a non-probability sampling method, convenience, or snowball sampling was used, where the participants themselves disseminate and recruit other participants.

### 2.2. Instruments and Variables

The first part of the questionnaire collected data on sociodemographic (sex and age) and anthropometric (weight, height, and BMI) variables.The BMI categories were: underweight (<18.5 kg/m^2^), normal weight (<18.5–24.9 kg/m^2^), overweight level I (18.5–24.9 kg/m^2^), overweight level II (25.0–26.9 kg/m^2^), obese type I (27.0–29.9 kg/m^2^), obese type II (30–34.9 kg/m^2^), and obese type III (35.0–39.9 kg/m^2^) [23].

The 10-item Perceived Stress Scale (PSS-10) [24], the Spanish adaptation of which was validated by Remor in 2006 [25], was used to assess perceived stress. The PSS-10 measures perceived stress in the past month in response to unpredictability, overburden, and lack of control [26]. It is designed to measure subjective perceived stress, and includes six items that cover negative feelings and four that cover positive feelings [27,28]. The responses are rated using a five-point Likert-type scale. The items covering negative feelings are scored thus: 0 = never, 1 = almost never, 2 = occasionally, 3 = almost always, and 4 = always—this is reversed to obtain the scores for the items covering positive feelings [29,30]. The total score ranges between 0 and 40. In the absence of a consensus on the cut-off values [31], in this study, those used by Adamson et al. in 2020 were taken as the reference—low stress (0–13), moderate stress (14–26), and high stress (27–40) [26].

Frequency of use questionnaires are very extensive (>44 items) and specific. Therefore, a questionnaire on consumption frequency with 19 items was self-compiled. The foods included were grouped under cereals, vegetables, fruits, and so on, following the recommendations of the Mediterranean Diet Pyramid. In addition, foods that have been identified as ‘unhealthy’ were also added such as energy drinks, snacks, and so on.

The last part of the questionnaire comprised questions on the increase in food consumption during the period under study and on eating behavior (eating between meals).

### 2.3. Data Collection

The questionnaire was administered online via a free platform (Google Forms) [32]. Social networks such as WhatsApp, Twitter, Facebook, and Instagram were used for its dissemination. Responses were collected from 10 January to 10 February 2021, the period of the ‘third wave’. Completion of the questionnaire was voluntary.

### 2.4. Statistical Analysis

The data obtained from the different variables were represented descriptively by frequency and percentage, and the quantitative variables were expressed by mean and standard deviation or dispersion.

The Kolmogorov–Smirnov test was used to determine whether the data followed a normal distribution; if it did not, the non-parametric chi-square test was carried out. Significant differences between the different groups were evaluated, assuming a confidence level of 95%. Significance was established at an alpha level of 0.05.

The statistical treatment was carried out using the R-Commander program.

## 3. Results

### 3.1. Participant Characteristics

A total of 637 people responded to the questionnaire—74.9% women and 25.1% men. The age range with the highest representation in this sample was 41–50. With respect to BMI, 49.4% respondents were classified as normal-weight and 47.0% as overweight or obese (Table 1).

### 3.2. Perceived Stress Scale (PSS-10)

The results of PSS-10 showed that 16.3% (*n* = 104) had low stress, 76.9% (*n* = 490) had moderate stress, and 6.7% (*n* = 43) had high stress.

The female respondents presented a higher prevalence of high stress than their male counterparts (8.6 vs. 1.2). The younger respondents (≤20 years) presented a higher percentage of moderate stress, with those in the age range 21–30 reporting high stress (10.6%).

According to BMI, those with type I obesity had high stress (13.6%) and the normal-weight ones had low stress (20.6%) (Table 2).

Table 3 shows the results obtained for the items covering negative feelings—it can be seen that for item 6, the mean wasclose to 1 (almost never), and items 3 and 9 exceeded 2 (occasionally).

For the items covering positive feelings, the results were close to 2 (occasionally), although the mean for item 5 was somewhere between 2 and 3 (almost always) (Table 4).

### 3.3. Frequencies of Food Consumption

Figure 1 shows the frequency of consumption of foods with their daily recommendations. More than 50.0% of respondents consumed dairy products and derivatives (57.4%), coffee, tea, and cocoa (57.3%), fruits (55.6%), and vegetables (55.4%).

Legumes and nuts were the least frequently consumed foods on a daily basis, with 74.1% and 61.8%, respectively.

Coffee, tea, and cocoa (15.1%), fruits (14.1%), and dairy products and derivatives (13.2%) were the most frequently consumed foods daily (>thrice/day).

Amongst the foods that should be consumed weekly, chicken, turkey, and/or rabbit meat (50.4%), and eggs and derivatives (49.0%) were consumed >thrice/week. Fish and seafood (45.7%) were the most consumed (1–3 times/week). Energy drinks (93.1%), commercial juices (79.6%), and snacks and jellybeans (78.3%) were the least consumed weekly (Figure 2).

### 3.4. Changes in Food Consumption Frequency and Eating Behavior

As much as 36.1% (*n* = 230) of respondents reported that they had increased their food consumption frequency, while 14.1% (*n* = 90) responded that they might have increased this. Moreover, 32.2% (*n* = 205) associated this increase with perceived stress during the ‘third wave’.

The foods that showed the highest increase in consumption frequency amongst the respondents were fruits (13.0%), vegetables (11.3%), snacks and jellybeans (10.8%), and nuts (10.6%). Those with high stress had increased the consumption of nuts (16.3%), snacks and jellybeans (14.0%), and coffee, tea, cocoa, and soft drinks (9.3%).

During the ‘third wave’, 38.8% (*n* = 247) of respondents had increased the intake of food between meals. There were no differences between the different BMI classification categories. Furthermore, 41.5% of the female respondents had increased their intake of food between meals, as opposed to 30.6% of their male counterparts. Analyzing by age, it was observed that the younger respondents had increased their intake of food between meals the most (58.1%), with the older ones (≥71 years) increasing this intake the least. Moreover, increase in this intake was also observed among 57.8% of the respondents who reported an increase in food consumption frequency.

The increased intake of food between meals was most pronounced in those with high perceived stress scores on the PSS-10 (65.1%), followed by those with moderate stress (40.4%) and low stress (20.2%) (Table 5).

### 3.5. Weight Gain

Of the total, 46.8% (*n* = 298) of respondents reported that they had not gained weight since the start of the ‘third wave’, while 21.8% (*n* = 139) were unsure if they had; on the other hand, 19.3% (*n* = 123) responded that they had gained 1–2 kg, followed by 8.1% (*n* = 52) who expressed that they had gained 3–4 kg, 2.0% (*n* = 13) who reported a weight gain > 6 kg, and 1.9% (*n* = 12) who had gained 5–6 kg.

There were no statistically significant differences between sex, age, and weight loss. The overweight or obese respondents reported higher weight gain in percentage terms than the underweight and normal-weight ones. A total of 50.6% of the respondents with type II obesity gained between 1 and 4 kg, while 9.1% of those with type I obesity reported a weight gain >6 kg, which could lead to them being classified under type II or III obesity.

Those who reported an increase in food consumption frequency reported greater weight gain compared with those who had not increased theirfrequency. This finding was more pronounced in those who had increased between-meal intake. In a more detailed way, it was observed in 30.0% of the respondents who gained 1–2 kg and 15.4% who gained 3–4 kg. 65.0% of the subjects who increased their consumption of industrial bakery products gained weight, followed by 53.3% who increased their intake of snacks and jelly beans. Respondents who increased their consumption of fruit and vegetables were the subjects who gained the least weight (72.3% showed no weight gain).

Perceived stress was another factor that influenced weight gain, with the respondents with high stress gaining more weight than those who reported a 1–2 kg weight gain (Table 6).

## 4. Discussion

Since the onset of the COVID-19 pandemic, research has been conducted on how this period has affected the mental health of the population. There have not been studies to date exploring the stress perceived during the ‘third wave’. 

The results of this study show that a high percentage of respondents experienced moderate stress in January–February 2021, when the number of cases and deaths in Spain rose again following the relaxation of measures in the previous month (Christmas). Our numbers related to moderate and high stress (83.6%) werehigher than those obtained by other authors [33,34,35,36] who conducted their studies during the lockdown or immediately after it from March to May 2020. This difference may be due to the study population, but can be attributed to the characteristics of this new period in the pandemic. In this period, the number of cases and deaths, and the burden on healthcare centers increased considerably, besides the emergence of new variants of the virus (the British strain) that are more contagious.

Furthermore, high perceived stress was observed more amongst women and those aged between 21 and 30. This result wassimilar to those reported during the first phase of the pandemic by other researchers [33,37].

It has been almost a year since the beginning of the lockdown to the date of the current study, which could have influenced the respondents’ responses, although they reported almost always feeling nervous or stressed about the pandemic and being upset that things were out of their control, they also reported almost always suggesting that they often feel that things are going well and that they have everything under control in relation to the pandemic. It should not be forgotten that perceived stress is defined as a person’s understanding of the stress he or she is exposed to at a given time or in a specific period [36], and possible coping measures. If negative perception of coping increases, it may lead to anxiety and/or depression, and decreased quality of life including sleep problems [38].

Perceived stress interferes with eating behavior and can lead to increased food consumption [39]—this includes increased consumption, especially of foods high in calories and rich in sugar, fats, and salt [18,19,40]. In this study, 36.1% of respondents expressed that they had increased their frequency of consumption of some foods, while 14.1% were unsure whether they had done so. In addition, consumption of nuts, snacks, and jellybeans, along with coffee, tea, cocoa, and soft drinks, had increased the most—this finding has been reported by other studies conducted during the lockdown [41,42].

During the period studied, 38.8% of respondents had increased their intake between main meals. This percentage isslightly lower than that obtained by Kriaucioniene et al. in April 2020 (45.1%) [43] and by Sidor and Rzymskiduring the lockdown in Poland (52.0%) [44]. This increase in intake between meals was more prevalent in the female (41.5%) and young respondents (58.1%) as well as in those who reported an increase in food consumption frequency (57.8%). Moreover, it was more pronounced in those with high perceived stress score on the PSS-10 (65.1%), followed by those with moderate stress (40.4%) and low stress (20.2%).

The association between perceived stress or increased food consumption and increased body weight, which is a risk factor, was studied [45]. The overweight or obese respondents reported weight gain in a higher percentage than the underweight and normal-weight ones. A total of 50.6% of the respondents with type II obesity gained between 1 and 4 kg, while 9.1% of those with type I obesity reported a weight gain >6 kg, which could lead to them being classified under types II or III obesity. These results are consistent with those of other studies [44]. The respondents who reported an increase in food consumption frequency reported greater weight gain compared with those who had not increased this frequency; this was more pronounced in those who had increased eating between meals, as observed in 30.0% of the respondents who gained 1–2 kg and 15.4% who gained 3–4 kg. This may be due to the increase in consumption of foods high in calories, sugar, and fats. In addition, perceived stress was observed as another factor influencing weight gain, with the respondents having high stress gaining weight.

The results provided serve as a starting point for the implementation of health policies in the field of nutrition to help improve health literacy in this field. In addition, it is important for health care providers to advise the population on stress management and food consumption, especially those foods that are less healthy or whose consumption should be occasional.

Among the limitations of the study, the type of sampling used in this study may lead to selection bias, as the sample was obtained non-randomly. This bias has been assumed because access to the study population at the national level was very difficult due to the situation in which the country was immersed and the mobility restrictions at the time. The under-representation of men in relation to the population as a whole could lead to a gender bias. Another limitation present is the difference in representation by age, so results should be interpreted with caution.Furthermore, the use of an online questionnaire may lead to an acquiescence response bias, although the questionnaire was simplified to reduce response time. However, Ekman et al., in 2006, stated that the bias with the collection of information through web questionnaires was no greater than that caused by paper questionnaires [46].

The generalizability of the results is limited because of the non-probability sample and the biases above-mentioned. Moreover, as it is a voluntary questionnaire, it is possible that those subjects who weremore aware of the problem may have participated. Nevertheless, it should be noted that, although the present research cannot be representative of the entire Spanish population, it has achieved a broad coverage, being a starting point for future, more specific research.

## 5. Conclusions

During this “third wave”, perceived stress levels have been high.

The level of perceived stress increases the risk of increases food consumption such as increased intake, especially of foods high in calories, sugars, and fats, and more snacking. This leads to weight gain, and consequently, to an increase in the prevalence of obesity in the population.

Therefore, measures should be taken to reduce people’s perceived stress, to curb unhealthy behaviors such as increased food consumption, or measures on nutrition education should be carried out in periods of high stress.

## Figures and Tables

**Figure 1 nutrients-13-02380-f001:**
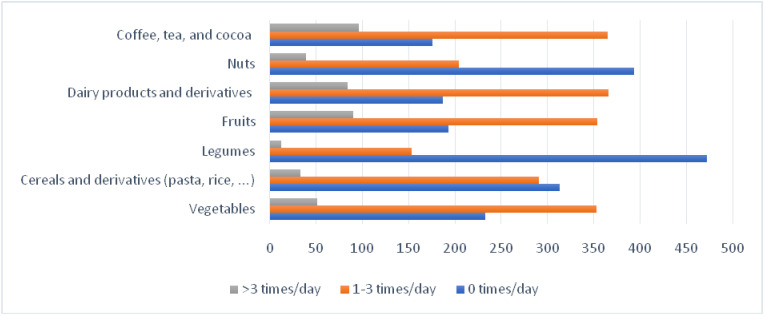
Frequency of consumption. Foods with daily recommendations.

**Figure 2 nutrients-13-02380-f002:**
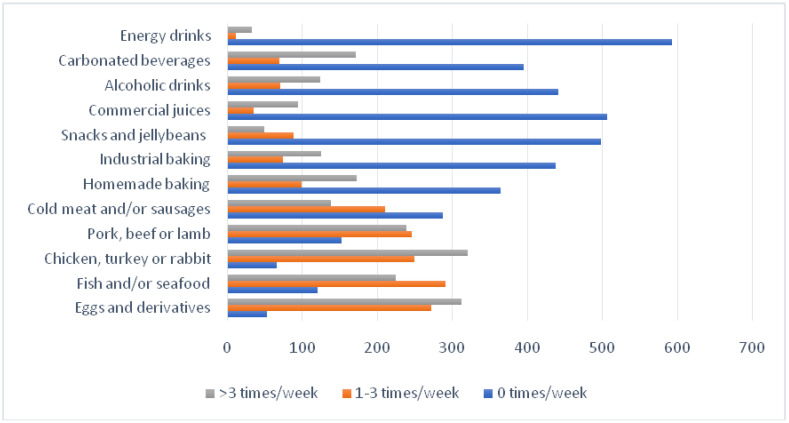
Frequency of consumption. Foods with weekly or occasional recommendations.

**Table 1 nutrients-13-02380-t001:** Sociodemographic and anthropometric variables.

	*n*	%	*p*
**Gender:**			<0.001 **
• Female	477	74.9	
• Male	160	25.1	
**Age:**			<0.001 **
• ≤20 years	31	4.9	
• 21–30 years	123	19.3	
• 31–40 years	178	27.9	
• 41–50 years	187	29.3	
• 51–60 years	87	13.6	
• 61–70 years	29	4.5	
• ≥71 years	2	0.3	
**BMI:**			<0.001 **
• Low weight	23	3.6	
• Normal weight	315	49.4	
• Overweightlevel I	101	15.8	
• Overweightlevel II	109	17.1	
• Obesitytype I	66	10.3	
• Obesitytype II	16	2.5	
• Obesity type III	7	1.1	

BMI: Body mass index; ** 0.001.

**Table 2 nutrients-13-02380-t002:** Assessment of perceived stress during the “third wave”.

	Low Stress	Moderate Stress	High Stress	*p*
	*n* (%)	*n* (%)	*n* (%)	
**Gender:**				<0.001 **
• Female	60 (12.6)	376 (78.8)	41 (8.6)	
• Male	44 (27.5)	114 (71.2)	2 (1.2)	
**Age:**				0.23
• ≤20 years	1 (3.2)	28 (90.3)	2 (6.5)	
• 21–30 years	18 (14.6)	92 (74.8)	13 (10.6)	
• 31–40 years	33 (18.5)	133 (74.7)	12 (6.7)	
• 41–50 years	32 (17.1)	140 (74.9)	15 (8.0)	
• 51–60 years	15 (17.2)	71 (81.6)	1 (1.1)	
• 61–70 years	5 (17.2)	24 (82.8)	0 (.0)	
• ≥71 years	0 (0.0)	2 (100.0)	0 (.0)	
**BMI:**				0.046 *
• Low weight	1 (4.3)	21 (91.3)	1 (4.3)	
• Normal weight	65 (20.6)	233 (74.0)	17 (5.4)	
• Overweightlevel I	11 (10.9)	86 (85.1)	4 (4.0)	
• Overweightlevel II	18 (16.5)	80 (73.4)	11 (10.1)	
• Obesitytype I	7 (10.6)	50 (75.8)	9 (13.6)	
• Obesitytype II	1 (6.2)	14 (87.5)	1 (6.2)	
• Obesitytype III	1 (14.3)	6 (85.7)	0 (.0)	

BMI: Body mass index; * 0.05; ** 0.001.

**Table 3 nutrients-13-02380-t003:** Items related to negative perceptions reported by respondents during the ‘third wave’ of the COVID-19 pandemic in Spain.

Items	Mean	SD	CI	D	*p*
**1. I feel something serious is going to happen in relation to the pandemic.**	1.72	0.99	1.64–1.80	0.255	<0.001 **
**2. I feel that things are out of my own control in relation to the pandemic.**	1.45	1.07	1.36–1.53	0.195	<0.001 **
**3. I feel nervous or stressed about the pandemic**	2.09	0.96	2.02–2.17	0.257	<0.001 **
**6. I feel unable to accomplish strategies for the prevention of Covid-19 disease**	1.26	1.05	1.18–1.35	0.206	<0.001 **
**9. I feel upset because things are not under control in relation to the pandemic.**	2.31	1.12	2.22–2.39	0.182	<0.001 **
**10. I feel I can’t cope with increasing difficulties related to the pandemic.**	1.69	1.10	1.60–1.77	0.172	<0.001 **

SD: Standard deviation; CI: Confidence interval; D: Kolmogorov–Smirnov value; ** 0.001.

**Table 4 nutrients-13-02380-t004:** Items related to positive perceptions reported by respondents during the ‘third wave’ of the COVID-19 pandemic in Spain.

Items	Mean	SD	CI	D	*p*
**4. I feel can manage by myself personal affairs in relation to the pandemic.**	1.68	1.03	1.60–1.76	0.204	<0.001 **
**5. I feel that things are going wellin relation to the pandemic.**	2.47	0.98	2.39–2.54	0.213	<0.001 **
**7. I feel I can cope with all difficulties related to the pandemic**	1.89	0.99	1.81–1.97	0.212	<0.001 **
**8. I feelIhave everything under control in relation to the pandemic.**	2.36	1.04	2.28–2.44	0.204	<0.001 **

SD: Standard deviation; CI: Confidence interval; D: Kolmogorov-Smirnov value; ** 0.001.

**Table 5 nutrients-13-02380-t005:** Assessment of between-meal intake during the ‘third wave’ of the pandemic in Spain.

	Increased between-Meal Intake (%)	Not Increased between-Meal Intake (%)	No between-Meal Intake at All (%)	*p*
**Gender:**				0.048 *
• Female	41.5	42.3	16.1	
• Male	30.6	51.2	18.1	
**Age:**				0.007 *
• ≤20 years	58.1	32.3	9.7	
• 21–30 years	47.2	38.2	14.6	
• 31–40 years	41	39.9	19.1	
• 41–50 years	35	50.3	13.9	
• 51–60 years	32.2	47.1	20.7	
• 61–70 years	10.3	65.5	24.1	
• ≥71 years	0	100	0	
**BMI:**				0.66
• Low weight	30.4	43.5	26.1	
• Normal weight	36.8	45.7	17.5	
• Overweight level I	37.6	43.6	18.8	
• Overweight level II	42.2	42.2	15.6	
• Obesity type I	40.9	47	12.1	
• Obesity type II	62.5	31.2	6.2	
• Obesity type III	42.9	57.2	0	
**Increased food intake:**				<0.001 **
• Yes	57.8	31.3	10.9	
• Maybe	47.8	42.4	10	
• No	22.4	54.9	22.7	
**Perceived stress:**				<0.001 **
• Low	20.2	55.8	24	
• Moderate	40.4	44.3	15.3	
• High	65.1	20.9	14	

BMI: Body mass index; * 0.05; ** 0.001.

**Table 6 nutrients-13-02380-t006:** Assessment of weight-gain during the ‘third wave’ of the pandemic in Spain.

	Weight Gain						
	No (%)	I’m not sure (%)	1–2 kg (%)	3–4 kg (%)	5–6 kg (%)	>6 kg (%)	*p*
**Gender:**							0.07
• Female	45.1	24.5	17.8	8.6	1.9	2.1	
• Male	51.9	13.8	22.8	6.9	1.9	1.9	
**Age:**							0.22
• ≤20 years	38.7	25.8	32.3	0	0	3.2	
• 21–30 years	39.8	26	14.6	13.8	3.3	2.4	
• 31–40 years	45.5	25.3	19.1	6.2	1.1	2.8	
• 41–50 years	52.9	18.2	17.1	8	3.2	0.5	
• 51–60 years	49.4	16.1	23	9.2	0	2.3	
• 61–70 years	44.8	17.2	31	3.4	0	3.4	
• ≥71 years	50	50	0	0	0	0	
**BMI:**							<0.001 **
• Low weight	60.9	26.1	8.7	4.3	0	0	
• Normal weight	48.6	26.7	17.5	6	1	0.3	
• Overweightlevel I	47.5	19.8	19.8	10.9	2	0	
• Overweightlevel II	44	12.8	25.7	8.3	4.6	4.6	
• Obesitytype I	37.9	21.2	19.7	9.1	3	9.1	
• Obesitytype II	43.8	0	25.6	25	0	6.2	
• Obesitytype III	42.9	12.3	14.3	28.6	0	0	
**Increased food intake:**							<0.001 **
• Yes	32.6	21.7	23.5	13.9	4.3	3.9	
• Maybe	32.2	26.7	27.8	7.8	1.1	4.4	
• No	61.2	20.5	13.9	4.1	0.3	0	
**Increased intake between meals:**							<0.001 **
• Yes	23.1	22.3	30	15.4	4.5	4.9	
• No	64.2	17	16	2.8	0	0	
**Not increased intake**	60.9	23.2	11.3	3.9	0.4	0.4	
**Perceived stress:**							<0.001 **
• Low	59.6	20.2	13.5	1.9	1.9	2.9	
• Moderate	45.9	22	19.6	10	1.2	1.2	
• High	25.6	23.3	30.2	2.3	9.3	9.3	

BMI: Body mass index; ** 0.001.

## Data Availability

The data are collected in a database prepared by the research team.

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
