# Peer review of "Perceived Stress and Increased Food Consumption during the ‘Third Wave’ of the COVID-19 Pandemic in Spain"

_nutrients, 2021, doi:10.3390/nu13072380_

Round 1
Reviewer 1 Report
Dear authors,
Your manuscript is interesting but I need you to answer some questions:
MATERIALS AND METHODS
Study design and participants:
- The authors say: “This was an observational, descriptive, and cross-sectional study…”. A cross-sectional study by definition is observational and descriptive. The information is redundant. I suggest saying that it is a "cross-sectional study" only.
- Body mass index has been a measure that has been called into question for many years. Today there are even apps that are better. For example the Body Volume Indicator (BVI). Authors must justify why they use such an imprecise measure.
- Have the authors used the classic form of BMI or some modified formula (e.g. Trefethenen, 2013)?
Data collection
- The authors must include the response rate of the participants in the study.
- What was the target population? How was the sample chosen? The authors must specify it.
- What have you done to avoid selection biases?
- What criteria were used to select the sample? It is not the same for those people who are confined or not; It is not the same if they were physically activity or are sedentary; It is not the same if they have suffered from covid-19 disease or not ...
Ethical considerations:
- Have you consulted the ethics committee? The authors must mention and say the reference.
DISCUSSION
- The authors have not explained how to apply the research to clinical practice.
- The authors have not included study limitations.
CONCLUSIONS
- In the first wave there was strict confinement and in the third wave there was not. Saying that perceived stress is higher is risky.
Author Response
Dear reviewer,
Firstly, we appreciate the time dedicated to our manuscript, as well as the clarifications you request, which help us to understand the doubts that a future reader may have, if the manuscript gets published.
Secondly, we answer to the questions that you have made, with aim of resolving doubts raised by our manuscript.
Study design and participants:
- The authors say: “This was an observational, descriptive, and cross-sectional study…”. A cross-sectional study by definition is observational and descriptive. The information is redundant. I suggest saying that it is a "cross-sectional study" only.
Modified as recommended.
- Body mass index has been a measure that has been called into question for many years. Today there are even apps that are better. For example the Body Volume Indicator (BVI). Authors must justify why they use such an imprecise measure.
Although BMI has been questioned as a measure, it is still used to categorise overweight and obesity. Perhaps the BVI is a more valid indicator because it takes into account the location of body fat mass. We do not have the data on waist circumference and waist-to-hip ratio to be able to do this. Also, as it is an online questionnaire, the calculation of waist circumference and hip circumference could be biased, but we will keep this in mind for future research relating BMI to health status.
- Have the authors used the classic form of BMI or some modified formula (e.g. Trefethenen, 2013)?
The BMI was calculated using the classic form and taking the parameters of the Spanish Obesity Society (SEEDO) as a reference for categorisation.
Data collection
- The authors must include the response rate of the participants in the study.
The response rate was 100, i.e. all subjects who answered the questionnaire did so in full. As this is an online questionnaire, we do not have a link between the dissemination of the questionnaire and the response rate. We have reviewed other studies that use this methodology and the response rate does not appear, perhaps because of this same problem.
- What was the target population? How was the sample chosen? The authors must specify it.
In section 2.1. Study design and participants, the study population appears as "adult Spanish population". The sample was obtained by convenience sampling through an online questionnaire.
- What have you done to avoid selection biases?
Dissemination via online means that all subjects have the same probability of being present in the sample, although there are always biases due to the type of sampling carried out.
- What criteria were used to select the sample? It is not the same for those people who are confined or not; It is not the same if they were physically activity or are sedentary; It is not the same if they have suffered from covid-19 disease or not ...
Convenience sampling is a technique used when there are no criteria to be considered for a person to be part of the sample, so they did not use accessory or specific criteria, except for age. Due to the anonymisation of the data, it is impossible to introduce these variables into the sample and obtain different subgroups, according to physical activity or confinement, but the authors will keep this in mind for future research with the aim of improving it.
Ethical considerations:
- Have you consulted the ethics committee?
The authors must mention and say the reference.
Approval was sought from the ethics committee but due to its low ethical burden it was not necessary to give an opinion. At the beginning of the questionnaire, the consent of the participants was requested in the following letter: "The questionnaire is completely anonymous, as the answers obtained cannot be related to a specific person, as it does not contain personal data (name, ID card number, etc.). The answers will be processed in compliance with Organic Law 3/2018, of 5 December, on Personal Data Protection and guarantee of digital rights".
DISCUSSION
- The authors have not explained how to apply the research to clinical practice.
These data do not influence clinical practice, but have implications for the health of society. A paragraph has been added to this point at the end of the discussion.
- The authors have not included study limitations.
A paragraph on the limitations of the study has been included.
CONCLUSIONS
- In the first wave there was strict confinement and in the third wave there was not. Saying that perceived stress is higher is risky.
The conclusions were amended.
In addition, following the recommendations of reviewer 2, the p-values of all tables have been modified, based on the article published by Cole (Cole TJ. Too many digits: the presentation of numerical data. Arch Dis Child. 2015;100(7):608-609. doi:10.1136/archdischild-2014-307149)
Once again, we appreciate the time and attention dedicated to our manuscript. We really hope we have reached your expectations, with the modifications made and that the explanations to those that we have not modified be considered as appropriate.
Kind regards.
Reviewer 2 Report
This manuscript examines the role of perceived stress due to the COVID-19 pandemic on food addition and weight change using data from 477 females and 160 males living in Spain. It found that consumption of some foods increased as did eating between meals. Respondents with higher stress reported greater weight gain. The results seem in line with what would be expected. However, attention should also be given to confounding factors such as reduced exercise during the lockdown periods. Also, would dietary shifts be due to reduced income due to the pandemic?
Participants were respondents to a questionnaire posted on social media. Perhaps the baseline characteristics of the respondents could be compared to that of the residents in Spain in general to see whether there might be some bias regarding the respondents. The female/male ratio is not similar to the population of Spain. Not sure about the BMI categories, and it would be useful if the BMI ranges were provided for each category.
|
Another aspect studied was whether perceived stress or food addiction increased |
272 |
|
body weight, which is a risk factor [45]. |
Comment: It is not clear what “which” refers to. The sentence should be revised.
Lockdowns also reduce time spent outdoors, therefore, the amount of exercise. Exercise affects risk of obesity as well. It might be worth discussing this point briefly. A search of pubmed.gov with “covid-19, lockdown, exercise, obesity” found 29 entries. A paragraph should be added outlining the strengths and limitations of this study. Not having data on exercise should be listed as a limitation. There may be other limitations.
The foods that shown the highest increase in consumption frequency amongst the 188
respondents were fruits (13.0%), vegetables (11.3%), snacks and jellybeans (10.8%), and 189
nuts (10.6%). Those with high stress had increased the consumption of nuts (16.3%), 190
snacks and jellybeans (14.0%), and coffee, tea, cocoa, and soft drinks (9.3%). 191
|
Those who reported an increase in food consumption frequency reported greater |
218 |
|
weight gain compared with those who had not increased this frequency. This finding was |
219 |
|
more pronounced in those who had increased between-meal intake. |
Comment: Is it possible to reach any conclusion on which of these intakes were more correlated with changes in weight? What is known about between-meal intake food groups? It might be the consumption for those with high stress.
Significant digits. The general rule is that no more non-zero digits should be given than are justified by the uncertainty of the value.
See "Too many digits: the presentation of numerical data"
https://www.ncbi.nlm.nih.gov/pmc/articles/PMC4483789/
If the uncertainty is greater than about 7%, only two non-zero digits are justified.
P values should be given to two decimal places unless the first two are 00 or the number lies between 0.045 and 0.050.
Thus, please check the p values such as in Table 2. .228 should be 0.23 while .046 should be 0.046. Be sure to add a “0” before any number less than 1.0
Please review all numbers in abstract, text, tables, and figures and adjust accordingly.
Author Response
Dear reviewer,
Firstly, we appreciate the time dedicated to our manuscript, as well as the clarifications you request, which help us to understand the doubts that a future reader may have, if the manuscript gets published.
Secondly, we answer to the questions that you have made, with aim of resolving doubts raised by our manuscript.
This manuscript examines the role of perceived stress due to the COVID-19 pandemic on food addition and weight change using data from 477 females and 160 males living in Spain. It found that consumption of some foods increased as did eating between meals. Respondents with higher stress reported greater weight gain. The results seem in line with what would be expected. However, attention should also be given to confounding factors such as reduced exercise during the lockdown periods. Also, would dietary shifts be due to reduced income due to the pandemic?
The data collection period was the 3rd wave, and in that period in Spain there was no lockdown, but restrictions on mobility, the use of masks and safety distances. In addition, vaccination of essential groups and people at risk (healthcare workers and institutionalised elderly people) began in the same period. Physical activity was allowed, albeit with slight restrictions.
Participants were respondents to a questionnaire posted on social media. Perhaps the baseline characteristics of the respondents could be compared to that of the residents in Spain in general to see whether there might be some bias regarding the respondents. The female/male ratio is not similar to the population of Spain. Not sure about the BMI categories, and it would be useful if the BMI ranges were provided for each category.
Convenience sampling provides the population with the same probability of being present in the sample. The cut-off points to describe the categories are those proposed by the Spanish Obesity Society (SEEDO) for the adult population. The cut-off points had not been added because it is a secondary variable, but it has been added in the manuscript.
Comment: It is not clear what “which” refers to. The sentence should be revised.
In the sentence "Another aspect studied was whether perceived stress or food addiction increased body weight, which is a risk factor", the first two aspects were risk factors for body weight gain.
Lockdowns also reduce time spent outdoors, therefore, the amount of exercise. Exercise affects risk of obesity as well. It might be worth discussing this point briefly. A search of pubmed.gov with “covid-19, lockdown, exercise, obesity” found 29 entries. A paragraph should be added outlining the strengths and limitations of this study. Not having data on exercise should be listed as a limitation. There may be other limitations.
As mentioned above, there was no confinement during this period and physical exercise could be performed without major limitations, except for a possible blockage of the locality, but it did not prevent movement through the locality. We understand that this situation did not alter the performance of the exercise in the subjects already performing it. In addition, a paragraph on limitations has been added at the end of the discussion that mentions this point.
Comment: Is it possible to reach any conclusion on which of these intakes were more correlated with changes in weight? What is known about between-meal intake food groups? It might be the consumption for those with high stress.
A paragraph on weight gain and food consumption has been added. It was not studied which foods were the most consumed between meals, only the increase in these types of meals was recorded.
Significant digits. The general rule is that no more non-zero digits should be given than are justified by the uncertainty of the value.https://www.ncbi.nlm.nih.gov/pmc/articles/PMC4483789/
Following their recommendations, the p-values in all tables have been modified.
Once again, we appreciate the time and attention dedicated to our manuscript. We really hope we have reached your expectations, with the modifications made and that the explanations to those that we have not modified be considered as appropriate.
Kind regards.
Reviewer 3 Report
Abstract: Please include the number of respondents included in the studies and specify what “questionnaire” is included in the survey together with the PSS-10.
Introduction: Although I consider the introduction well developed with respect to the rationale, I suggest a revision of the English because some sentences are difficult to understand.
Moreover, I suggest including the hypothesis associated to the aim of the study, according to the previous literature.
Methods: In Participants or data collection pleas include the inclusion and exclusion criteria of the participants in the sample, as well as the control strategies adopted to reduce the errors.
Results: In frequencies table there are some problems of approximation in the percentages’data (e.g., male, 31-40 years, BMI low weight)
Discussion: in the limit section the retrospective nature of the survey (on changes in eating habits and weight condition) should be included.
Author Response
Dear reviewer,
before we begin we would like to thank you for taking the time to review our manuscript, which will help us to improve its quality.
In the following, we try to respond to the comments made in your review, hoping to meet your expectations and clarifying all the points that, in your opinion, should be modified.
Summary: Please include the number of respondents included in the studies and specify which "questionnaire" is included in the survey along with the PSS-10.
Add these suggestions in the abstract of the manuscript.
Introduction: Although I consider the introduction to be well developed with respect to the rationale, I suggest a revision of the English because some sentences are difficult to understand.
The entire text has been extensively revised and an official translator and English health professionals have been asked to help.
In addition, I suggest including the hypothesis associated with the objective of the study, in accordance with the previous bibliography.
The hypothesis under study is added and the term "Spanish population" is removed because the results are not generalisable due to the type of sampling used.
Methods: In Participants or reasons for data collection, the criteria for inclusion and exclusion of participants in the sample are included, as well as the control strategies adopted to reduce errors.
Section 2.1. The study design and participants have been modified.
Results: In the frequency table there are some approximation problems in the percentage data (e.g. male, 31-40 years old, BMI underweight).
We do not see the comment it conveys. All tables have been checked again to correct the existing errors.
Discussion: the retrospective nature of the survey (on changes in eating habits and weight status) should be included in the boundary section.
A new paragraph has been added to the discussion on limitations.
Again, we thank you for your time and willingness to review our manuscript.
We hope we have met your expectations on this occasion.
Kind regards.
Round 2
Reviewer 1 Report
Dear authors,
I think that his answers are not satisfactory. Methodologically, the work is not reproducible and has important errors:
MATERIALS AND METHODS
Data collection:
- If you say that the target population is the "adults Spanish population" the response rate cannot be 100%. If you do convenience sampling, you should know who sent the online questionnaire to (total population). What you do wrong in other investigations is no excuse for you to proceed in the same way.
- You say that you have followed the SEEDO classification. What are the requirements for the inclusion of subjects in your research? They must specify them properly.
- If you have done convenience sampling, you can put measures to avoid biases. Please specify them. You have not answered the question.
- What you say is not true. In a convenience sampling, criteria can be added to select the sample. Precisely the SEEDO says in its 2016 consensus the following:
"The BMI does not report the distribution of body fat, it does not differentiate between lean mass (MM) and MG, and it is a poor indicator in subjects of short stature, advanced age, muscular, with hydro saline retention or pregnant women ". Also, the consensus of the SEEDO 2020 and SEMERGEN specifically speaks of exclusion criteria for subjects and also says the following: "in individuals with a BMI between 25 and 35 kg / m2, the abdominal circumference must also be measured and thus discriminate between obesity central and peripheral ".
Therefore you should have taken this into account to avoid selection biases.
Ethical considerations:
- The data protection law is one thing and that a bioethics committee considers appropriate its way of proceeding in research is another. Although the data is anonymous, you have an obligation to save and safeguard that data. But if an ethics committee does not consider your research adequate, you cannot carry it out even if you respect the Data Protection Law.
DISCUSSION
- Community Health is implications for clinical practice. You can call them whatever you like.
Best regards
Author Response
Dear reviewer,
before we begin we would like to thank you for giving us another opportunity to try to respond to your clarifications. In addition, we believe that we did not express ourselves adequately or did not know how to express what we wanted to say and this has led you to think that the manuscript should not be published.
In the following, we try to respond to the comments made in your review, hoping to meet your expectations and clarifying all the points that you feel should be modified.
MATERIALS AND METHODS
Data collection:
If you say that the target population is the "adults Spanish population" the response rate cannot be 100%. If you do convenience sampling, you should know who sent the online questionnaire to (total population). What you do wrong in other investigations is no excuse for you to proceed in the same way.
We are sorry if our words made you think that we justified our errors with possible errors in other research. The authors intended to express the characteristics of convenience sampling. Section 2.1. Study design and participants has been modified, paying attention to the type of sampling. In the discussion, a point on limitations has been added where the limitations of using this type of sampling are also expressed.
In addition, the term "Spanish population” has been removed from the title and objective, as due to the type of sampling the results cannot be generalised.
You say that you have followed the SEEDO classification. What are the requirements for the inclusion of subjects in your research? They must specify them properly.
As this was a convenience or snowball sample, the participants themselves invited other participants. As mentioned, this may lead to some bias and is therefore mentioned in the limitations. The SEEDO classification was used to further classify the results obtained in the study, but not as an inclusion criterion.
If you have done convenience sampling, you can put measures to avoid biases. Please specify them. You have not answered the question.
The main bias of this type of sampling is the impossibility of generalising the data. It could not be controlled because we did not have access to data from the entire Spanish population. This limitation related to this bias has been mentioned in the paragraph on limitations.
What you say is not true. In a convenience sampling, criteria can be added to select the sample. Precisely the SEEDO says in its 2016 consensus the following: "The BMI does not report the distribution of body fat, it does not differentiate between lean mass (MM) and MG, and it is a poor indicator in subjects of short stature, advanced age, muscular, with hydro saline retention or pregnant women ". Also, the consensus of the SEEDO 2020 and SEMERGEN specifically speaks of exclusion criteria for subjects and also says the following: "in individuals with a BMI between 25 and 35 kg / m2, the abdominal circumference must also be measured and thus discriminate between obesity central and peripheral ".
Therefore you should have taken this into account to avoid selection biases.
The SEEDO classification was not an inclusion criterion, but a categorisation of the data. The aim of the study was not to discriminate between central and peripheral obesity, since, as we mentioned, this is not an inclusion criterion but a categorisation of results. With this categorisation we aimed to classify the weight status of the subjects who completed the questionnaire. Therefore, although the study may have selection biases due to the type of sampling used, the SEEDO classification has not given rise to this bias, because it is not a criterion for selecting subjects.
Ethical considerations:
The data protection law is one thing and that a bioethics committee considers appropriate its way of proceeding in research is another. Although the data is anonymous, you have an obligation to save and safeguard that data. But if an ethics committee does not consider your research adequate, you cannot carry it out even if you respect the Data Protection Law.
We do not claim that an ethics committee should not be called upon to assess our project before we start it. We mentioned that due to the low ethical burden and the anonymisation of the data, the committee to which we presented our objective and methodology made it clear that the study could be carried out without requesting a review by the committee.
DISCUSSION
Community Health is implications for clinical practice. You can call them whatever you like.
The authors did not mean to imply that community health has no implications for clinical practice. The lead author is a clinician and believes that community health is a fundamental pillar of population health. In the discussion we mention that governments should push for policies to improve nutritional health and stress management.
We appreciate all the time you have taken to review our manuscript and the comments you have sent us, as we believe this will improve it.
We apologise if we did not express our ideas in our responses to previous reviews and hope that we have met your expectations on this occasion.
Kind regards.